# Impact of Different Biochars on Microbial Community Structure in the Rhizospheric Soil of Rice Grown in Albic Soil

**DOI:** 10.3390/molecules26164783

**Published:** 2021-08-07

**Authors:** Dawei Yin, Hongyu Li, Haize Wang, Xiaohong Guo, Zhihui Wang, Yandong Lv, Guohua Ding, Liang Jin, Yu Lan

**Affiliations:** 1College of Agricultural Science, Heilongjiang Bayi Agricultural University, Daqing 163319, China; yindazhiyindawei@126.com (D.Y.); ndrice@163.com (H.L.); whzhlj@163.com (H.W.); guoxh1980@163.com (X.G.); byndwzh@163.com (Z.W.); Lvyd_1978@163.com (Y.L.); 2College of Cultivation and Cultivation, Heilongjiang Academy of Agricultural Sciences, Harbin 150000, China; hucheng229@126.com; 3College of Plant Nutrition and Resources, Beijing Academy of Agriculture and Forestry Sciences, Beijing 100097, China; 4College of Agricultural Science, Shenyang Agricultural University, Shenyang 110161, China

**Keywords:** biochar, albic soil, bacteria, fungi, community structure

## Abstract

The purpose of this study was to clarify the effects of biochar on the diversity of bacteria and fungi in the rice root zone and to reveal the changes in soil microbial community structure in the root zone after biochar application to provide a scientific basis for the improvement of albic soil. Rice and corn stalk biochar were mixed with albic soil in a pot experiment. Soil samples were collected at the rice maturity stage, soil nutrients were determined, and genomic DNA was extracted. The library was established using polymerase chain reaction (PCR) amplification. The abundance, diversity index, and community structure of the soil bacterial 16SrRNA gene V3 + V4 region and the fungal internal transcribed spacer-1 (ITS1) region were analyzed using Illumina second-generation high-throughput sequencing technology on the MiSeq platform with related bioinformatics. The results revealed that the biochar increased the soil nutrient content of albic soil. The bacteria ACE indexes of treatments of rice straw biochar (SD) and corn straw biochar (SY) were increased by 3.10% and 2.06%, respectively, and the fungi ACE and Chao indices of SD were increased by 7.86% and 14.16%, respectively, compared to conventional control treatment with no biochar (SBCK). The numbers of bacterial and fungal operational taxonomic units (OUT) in SD and SY were increased, respectively, compared to that of SBCK. The relationship between soil bacteria and fungi in the biochar-treated groups was stronger than that in the SBCK. The bacterial and fungal populations were correlated with soil nutrients, which suggested that the impacts of biochar on the soil bacteria and fungi community were indirectly driven by alternation of soil nutrient characteristics. The addition of two types of biochar altered the soil microbial community structure and the effect of rice straw biochar treatment on SD was more pronounced. This study aimed to provide a reference and basic understanding for albic soil improvement by biochar, with good application prospects.

## 1. Introduction

Albic soil is a low-yielding soil that possesses obstacles and exists widely throughout the world. There are 32 countries or regions throughout the world that exhibit similar distributions of albic soil, and the total area of albic soil in China is approximately 5.273 million ha [1]. Due to the severe problems caused by dense physical structure, poor nutrient content, and low biological activity, albic soil has been characterized as a low-yielding soil [1,2]. It is therefore of great strategic significance to improve low-yielding albic soil to ensure food security.

Microorganisms can improve soil fertility and productivity by participating in the degradation of organic matter, the formation of humus, the transformation of nutrients, and the cycles of carbon, nitrogen, phosphorus, and other elements [3,4]. The microbial community structure is closely related to soil quality and is extremely sensitive to changes in the living environment [5]. Microbial diversity and community structure can be used as important indicators for assessing soil health status [6]. Plant-growth-promoting rhizobacteria (PGPR) are considered beneficial microorganisms to be used as biofertilizer to promote plant growth and crop yield [7]. The PGPR in the soil act as biochemists, which can influence soil pH, contribute to plant biomass, increase enzyme activity, and improve mineralization (C, N) processes [8]. Various genera of rhizobacteria such as *Pseudomonas*, *Azospirillum*, *Azotobacter*, *Klebsiella*, *Enterobacter*, *Alcaligenes*, *Arthrobacter*, *Burkholderia*, *Bacillus,* and *Serratia* have been investigated to improve the growth and productivity of various plants [9,10].

Biochar is the carbon-rich product obtained from the thermochemical conversion of biomass under oxygen-limited conditions [11,12,13]. Many researchers have reported that biochar contains high levels of carbon and has a structure with physicochemical properties, including a well-developed pore structure, huge surface area, high degree of stability, great adsorption properties, and abundant nutrients that are beneficial to crop yield [14]. Biochar such as rice straw biochar can directly import N, P, K, and Si into this soil [15,16]. The corn biochar reduced the soil bulk density and improved the soil pH, total porosity, aggregate stability, and maximum field capacity during the critical period of soybean growth with a high water requirement [14]. Biochar acts as a conditioner to enhance soil physicochemical properties, a nutrient source for plants, and a carbon source, and a suitable environment for PGPR and indigenous microbes to enhance their growth promotion activities [17].

The biochar provides a good “shelter” for the habitat and for reproduction of microorganisms, and it also reduces survival competition among microorganisms [18,19,20]. The biochar affects soil microbial growth, diversity, and community compositions by directly providing growth promoters for soil biota or indirectly changing soil basic properties. The porous structure, labile C, high pH, and electrochemical properties of biochar play an important role in determining soil microbial abundance and communities and their mediated N and P cycling processes [19]. However, some studies have indicated that biochar has no effect on the overall microbial community structure [21]. The effects of biochar on microbial community structure are related to the properties of biochar and also to climatic conditions, crops, soil texture, and soil moisture [22].

The effects of biochar on soil microbial community structure have been studied in various soil types [21,23,24]. However, there are few reports examining the effect of biochar on the microbial community structure of low-yielding albic soil in Northeast China. Improving the microbial community structure of low-yielding albic soil is of great significance for increasing soil fertility. In this study, rice and corn straw biochar were selected for a pot experiment, and high-throughput sequencing technology was used. The effects of biochar on the diversity of bacteria and fungi in the rice root zone were investigated to reveal the changes in soil microbial community structure in the root zone after application of biochar to ultimately provide a theoretical basis for the improvement of low-yielding albic soil.

## 2. Results and Discussion

### 2.1. Influence of Biochar on the Nutrient Content of Albic Soil

The abundant organic carbon and minerals in biochar are also beneficial for increasing soil organic carbon content [15]. As presented in Table 1, SD and SY increased their soil organic matter content at the rice maturity stage by 1.05% and 1.31%, respectively, compared to that of SBCK. This may be due to the ability of biochar to adsorb unstable organic carbon through its surface, inhibit organic carbon mineralization, and promote the polymerization of adsorbed organic molecules to form organic matter [15]. Biochar contains a high carbon content (50.6%) and thus directly increases the soil organic matter content when applied to soil [2]. In this study, the total C content of corn stalk biochar was 50.60%, and the total C content of rice straw biochar was 51.41%. Moreover, biochar can release dissolved organic carbon (OC) [15].

Biochar application has been shown to significantly improve chemical properties of problem soils. The previous findings strongly indicate that biochar amendment altered the chemical properties, such as soil CEC, pH, soil TN, and AK contents [14,16,25]. Biochar applications can enhance the status of nutrients, especially N [26]. Some previous studies have noted the positive effects of biochar on soil inorganic N [27,28]. In this study, the content of alkali-hydrolyzed nitrogen (AN) in SD and SY was increased by 12.87% and 7.45%, respectively, compared to that in SBCK. The total nitrogen content (TN) of SD and SY was increased by 4.87% and 4.33%, respectively, compared to that of SBCK. This may be related to the following reasons. The biochar addition stimulated the abundance and growth of soil N-fixing bacteria [24], thereby increasing soil N nitrification and soil NH_4_^+^ content [28]. Moreover, the increased soil NO_3_^−^ from the biochar addition may have occurred because the biochar promoted soil N mineralization and decreased soil N leaching [26].

In our study, the content of available phosphorus (AP) in SD and SY was increased by 9.21% and 8.62%, respectively, compared to that of SBCK. The total phosphorus (TP) content of SD and SY was increased by 3.19% and 8.63%, respectively, compared to that of SBCK. The total potassium (TK) content of SD and SY was increased by 0.38% and 5.38%, respectively, compared to that of SBCK. The improvement in soil fertility was partly dependent on the nutrient supply obtained from the biochar addition [29]. Moreover, biochar possesses a rich pore structure and exhibits strong adsorption performance, and it can store nutrients and release nutrients slowly in the soil [30]. Additionally, the improved soil properties (e.g., the increased aggregation capacity and water storage capacity), which were favorable for increasing nutrient contents and decreasing nutrient leaching [31] also led to elevated soil fertility. The improved microbial activity after the biochar addition could accelerate nutrient release to the soil and elevate the nutrient content [32].

The results revealed that the application of rice straw biochar (SD) could increase the soil available potassium (AK) content; however, corn straw biochar (SY) exerted a certain inhibitory effect on the soil AK content, thus indicating that the effects of biochar derived from different sources on the soil AK content were different.

### 2.2. Depth Assessment of Soil Sample Sequencing

Using Illumina high-throughput sequencing, nine fungal samples were sequenced, and 684,677 pairs of reads were obtained. A total of 624,238 clean tags were generated after double-ended reads assembly and filtration, and each sample generated at least 55,249 clean tags. The average number of clean tags was 69,360. A total of 589,679 pairs of reads were obtained by sequencing nine bacterial samples. A total of 506,522 clean tags were generated after stitching and filtering of double-ended reads, and each sample generated at least 37,163 clean tags with an average of 56,280 clean tags.

Sequencing sequences were randomly selected, and a curve was constructed by combining the number of sequences extracted with the number of OTUs that they could represent [18]. The OTU was clustered at the 97% similarity level, and the dilution curves of each sample were generated (Figure 1). As presented in Figure 1, the curves of the bacteria and fungi gradually became flattened, thus indicating a reasonable sequencing quantity.

### 2.3. Effects of Biochar on the Abundance and Diversity of Soil Bacterial and Fungal Communities

Changes in soil microbial abundance as a result of biochar addition have been an area of great concern. The ACE and Chao indices reflect the species abundance of the community [33,34]. In this study, as presented in Table 2, biochar increased the ACE and Chao indices of soil bacteria. Compared to SBCK, the ACE indices of SD and SY of bacteria were increased by 3.10% and 2.06%, and the Chao indexes of SD and SY of bacteria were increased by 3.25% and 2.26%, respectively. The results revealed that the fungi ACE and Chao indices of SD were increased by 7.86% and 14.16%, respectively, compared to SBCK. This may be due to the abundant pore structure of biochar itself that provides a good “shelter” for the habitat and the reproduction of microorganisms by protecting them from adverse effects such as invasion and dehydration while at the same time reducing the survival competition among microorganisms [33]. Biochar amendment increases soil microbial abundance largely due to a direct result of the utilization of labile C in biochar and a small amount of nutrients by soil microorganisms [34]. The albic soil in this experiment is poor in nutrients and compact in physical structure. Biochar can indirectly affect the composition of the soil microbial community by improving soil properties, such as the changes of soil pH, adhesion, etc. [34].

However, different types of straw biochar had different effects on fungal abundance. The corn straw biochar (SY) reduced fungal abundance, the SY reduced the ACE index and Chao index of fungi to levels that were 5.63% and 2.45% lower, respectively, than that of SBCK. This may be due to the alkaline nature of biochar, as an increase in soil pH reduces the abundance of fungi [21]. The pH values of rice straw biochar and corn straw biochar adopted in this study are 7.87 and 8.50 respectively. The high pH value of corn straw biochar may inhibit the abundance of fungal community in albic soil. The corn straw biochar is rich in mineral elements and sufficient nutrients and can adversely affect the growth of mycorrhiza, and high salt content may inhibit the growth and reproduction of mycorrhiza in the soil [23]. In addition, the corn straw biochar may not provide a sufficient number of pores that are suitable for fungal survival [19].

Previous studies have indicated that biochar amendment differs in its effects on bacterial and fungal diversity. For example, Shuailin Li et al. (2020) reported that bacterial and fungal alpha diversity hardly changed after two years of addition of biochar alone or combined with N fertilizer. Hu et al. (2014) reported that bacterial diversity increased but fungal diversity decreased with short-term biochar addition in a red oxidized loam soil [35]. In this study, biochar had no significant effect on bacterial community diversity. The SD increased fungal community diversity, but SY decreased fungal community diversity. The Shannon index of fungi treated with SD was increased by 3.84% compared to SBCK. However, the Shannon index of fungi in SY was decreased by 6.99% compared to SBCK. This may be due to the fact that the diversity of microorganisms is jointly determined by species richness and evenness, and this may vary owing to the proportion of richness and evenness [36].

Different materials of biochar possess certain differences in composition and structure that exert different effects on soil physical and chemical properties and thus exert different effects on microbial community structure diversity [37].

The addition of two types of biochar altered the soil microbial communities, and the effect of rice straw biochar treatment on SD was more pronounced. This may be due to the rice straw biochar having better adaptability to the microbial population in albic soil types. Therefore, in the process of recycling agricultural waste into biochar, the original crops under specific soil types should be given priority.

### 2.4. Effects of Biochar on the Relative Abundance of Bacteria and Fungi in Root Zone Soil

Many studies have demonstrated that biochar has an impact on bacterial and/or fungal community compositions on short- or long-term scales [33,38]. However, the changes in community composition caused by the biochar remain unclear. Figure 2 presents the bar chart for soil bacteria and fungi at the level of phylum classification. According to Figure 2A, the bacteria present in soil samples include *Proteobacteria*, *Acidobacteria*, *Chloroflexi*, *Verrucomicrobia*, *Bacteroidetes*, *Gemmatimonadetes,* and other bacteria groups. *Proteobacteria*, *Acidobacteria*, *Chloroflexi*, and *Verrucomicrobia* exhibit relatively high abundances of 30.69%~34.97%, 22.20%~24.02%, 9.04%~11.68%, and 8.64%~10.07%, while all other types of bacteria exhibit a total abundance of 74.46%~76.40%.

As presented in Figure 2A, biochar increased the relative abundance of *Proteobacteria*, *Verrucomicrobia*, and *Bacteroidetes*. *Proteobacteria* are the largest group of bacteria. Compared to that of SBCK, the relative abundance *of Proteobacteria* in SD and SY was increased by 13.95% and 9.94%, respectively. *Proteobacteria* accounted for the largest proportion in the soil in terms of community composition and relative abundance, and this is consistent with the conclusions of previous studies [39,40]. This is due to the fact that the *Proteobacteria* is a eutrophic bacterium [41], and biochar application has been shown to significantly improve nutrient properties of albic soils (Table 1), leading to an increase in the abundance of *Proteobacteria*.

The relative abundance of *Verrucomicrobia* in SD and SY was increased by 16.57% and 8.46% compared to SBCK. Biochar may increase the relative abundance of *Verrucomicrobia* by increasing soil carbon content and adjusting soil pH [42]. In this study, SD and SY increased soil organic matter content by 1.05% and 1.31%.

The relative abundance of *Bacteroidetes* in SD and SY was increased by 0.39% and 23.62%, compared to SBCK. *Bacteroidetes* is closely associated with the conversion of organic materials such as DNA, proteins, and lipids [42]. As a carbon source, the application of biochar to soil is conducive to the improvement of the relative abundance of *Bacteroidetes* within the soil.

However, biochar reduced the relative abundance of *Acidobacteria*, *Chloroflexi*, *Gemmatimonadetes*, *Actinobacteria*, and *Armatimonadetes*. The relative abundance of *Acidobacteria* in SD and SY was decreased by 7.08% and 7.58%, compared to SBCK. Most *Acidobacteria* are acidophilic bacteria, and their abundance is negatively correlated with soil pH. In an environment possessing a low soil pH, the abundance of *Acidobacteria* is the highest [43]. Biochar is alkaline, and biochar can reduce the relative abundance of *Acidobacteria* by regulating soil pH values. In this study, the albic soil pH value was 5.35, and those of corn stalk biochar and rice straw biochar were 8.50 and 7.87.

*Acidobacteria* and *Chloroflexi* belong to the oligotrophic microorganism group [44,45], and their growth rate is inhibited and their relative abundance is reduced in the eutrophic environment of biochar. The relative abundance of *Chloroflexi* in SD and SY was decreased by 22.60% and 21.68%, compared to SBCK. *Chloroflexi* species are usually predicted to degrade plant compounds with pathways commonly identified for the degradation of cellulose, starch, and long-chain sugars and compete for labile carbon with other organisms [46]. Therefore, the lower relative abundance of *Chloroflexi* in SD and SY may limit the organic material degradation rate.

The relative abundance of *Actinobacteria* in SD and SY were decreased. *Actinobacteria*, as Gram-positive bacteria, play a vital role in organic matter turnover, including the decomposition of cellulose and chitin [47]. The decreased populations of *Actinobacteria* in SD and SY may symmetrically retard microbial organic matter decomposition, which partly drives the higher soil CO_2_ emissions [47].

As presented in Figure 2B, fungi present in the soil samples include *Ascomycota*, *Basidiomycota*, *Rozellomycota*, *Mortierellomycota*, *Chytridiomycota,* and other fungi. In addition to other fungi whose species have not been determined, the relative abundance of *Ascomycota* and *Basidiomycota* were 26.19%–58.19% and 2.19%–4.65%, respectively, and these belonged to the dominant fungi with a cumulative sum of 29.17%–62.84%.

Biochar increased the relative abundance of *Rozellomycota*, *Chytridiomycota*, *Apheliidiomycota,* and unclassified fungi. This increase in fungal species led to increased competition for energy and nutrients among species [48], resulting in a decrease in the relative abundance of major fungal groups and thus further demonstrated that biochar enriched the community structure of soil fungi.

However, biochar decreased the relative abundance of *Ascomycota*, *Basidiomycota*, and *Mortierellomycota*. *Basidiomycota* can roughly be divided into the following, saprobic fungi, which decompose organic matter; symbiotic fungi, which form a mutualistic relationship with other organisms; pathogenic or parasitic fungi, which infect plants, animals, and even other fungi [49]. The relative abundance of *Basidiomycota* in SD and SY was decreased by 43.22% and 23.46%, respectively, compared to SBCK. Mineralizable C has been reported to significantly decrease the relative abundance of *Basidiomycota*, and most fungal OTUs from *Basidiomycota* were assigned as nonsaprotrophs. As a microbial C source, the DOC probably promotes saprotroph growth and enhances their competitive capacity, leading to an overall decrease in diversity and a decline in fungal pathogens [50]. Hence, the decreased relative abundance of *Basidiomycota* in biochar treatments may be due to biochar increasing the organic matter (Table 1).

Figure 3 presents the bar chart for soil bacteria and fungi at the level of genus classification. As presented in Figure 3A, at the genus level of soil bacteria biochar increased the relative abundance of *Sphingomonas*, *uncultured_bacterium_f_DA101_soil_group*, *Geobacter,* and *uncultured_bacterium_f_Gemmatimonadaceae*. However, the relative abundances of *uncultured_bacterium_f_Acidobacteriaceae_[Subgroup_1]*, *Candidatus_Solibacter*, *Anaeromyxobacter*, *uncultured_bacterium_f_Anaerolineaceae*, *Anaerolinea,* and *Gemmatimonas* were decreased. *Bradyrhizobium* is widely regarded as a nitrogen-fixing and denitrification bacterium and participates in the nitrogen cycle [51]. In this study, the effect of different kinds of biochar on *Bradyrhizobium* was inconsistent. The corn stalk biochar (SY) increased the abundance of *Bradyrhizobium* and rice stalk biochar (SD) decreased. However, the effect of biochar on other PGPR was not detected. The biochar not only improves the physicochemical properties of the soil but also serves as carrier material for PGPR inoculation [9,52], and biochar provides a safer environment against various biological competitors in soil due to porous structure [53,54]. Further research is needed to confirm the biochar with PGPR efficacy in the long-term with respect to albic soil conditions.

As presented in Figure 3B, biochar increased the relative abundance of *Cladosporium* and *Didymella* at the genus level of soil fungi. However, the relative abundance of *Pseudeurotium*, *Alternaria*, *Mortierella*, *Solicoccozyma*, *Oidiodendron,* and *Neobulgaria* were decreased.

*Mortierella* was shown to successfully suppress the occurrence of clubroot disease (caused by Sclerotinia sclerotiorum) [55], which includes fast-growing saprobic fungi that mainly utilize simple soluble substrates and is associated with high cellulose content in the soil [56]. The relative abundances of *Mortierella* were positively correlated with soil TC content [57]. Yao et al. (2017) showed that biochar had no significant effect on *Mortierella* abundance. This study showed that biochar reduced the abundance of *Mortierella*. Although the biochar provided a carbon source for albic soil, it is speculated that the dominant fungi genera in the soil compete for carbon sources, leading to the decrease in *Mortierella* abundance.

*Alternaria* is one of the important fungal groups that can cause plant diseases, and nearly 90% of the reported species of *Alternaria* fungi in the world can cause field and postnatal losses [58]. The abundance of *Alternaria* could be reduced by applying cotton stalk biochar in soil not polluted by cadmium [55]. The experiment showed that the abundance of *Alternaria* was decreased by biochar, and the reduced abundance means that biochar application may be beneficial to the control of crop diseases. The mechanism of biochar inhibition of plant diseases is very complex, including direct inhibition of pathogen growth [59].

Significance analysis of inter-group differences revealed that when an LDA value of >4 was set, there were few markers with significant differences between different groups of samples. When an LDA value of >3 was set, the significant difference markers in soil bacteria were enriched in the SD and SY treatments, and the significant difference markers in SD and SY were 3, while that of SBCK was 0. Significant difference markers in soil fungi were enriched in SBCK and SD treatments, of which nine markers were found for SBCK and two markers were found for SD. This demonstrates that biochar affects the composition of bacteria and fungi within the soil (Figure 4).

### 2.5. Analysis of Soil Bacteria and Fungi Groups in the Context of Biochar Treatment

The composition and relative abundance of bacteria and fungi in saline-sodic soil changed after 3 years of applying different dosages of biochar under rice cultivation. The addition of biochar had no significant impact on the bacterial diversity, but the diversity of fungi showed a tendency to decrease [60]. The long-term influence of biochar addition on the fungal community was shown at the genus and OTU levels but not at the phylum level [48]. As indicated in Table 3, biochar increased the number of bacterial OTUs. The numbers of bacterial OTU units in SD and SY were 1549.67 and 1546.67, respectively, and these were increased by 3.01% and 2.81%, respectively, compared to the value of 1504.33 in SBCK. Biochar increased the number of fungal OUT units, and the number of fungal OUT units in SD and SY were 288.33 and 251.67, respectively. These were increased by 19.97% and 4.72% compared to the value of 240.33 in SBCK.

Venn diagrams can reflect the number of common and unique OTUs between groups or samples and can intuitively indicate the overlap of OTUs between groups or samples [61]. Apple biochar at 0.5%–4% dosage could significantly increase the number of unique bacteria OTUs and affect the bacteria groups [62], while the number of unique bacterial OTU was one to three times the common OTU numbers, but they had no significant effect on the fungal groups [48]. As presented in Figure 5A, the number of common bacterial OTUs among different treatments was 1624, the number of unique bacterial OTUs without biochar application (SBCK) was 1, and the numbers of unique bacterial OTUs in biochar treatment of SD and SY were 4 and 3, respectively. The number of common fungal OTUs among the different treatments was 280, the number of unique fungal OTUs in the absence of biochar (SBCK) was 25, and the numbers of unique fungal OTUs in the biochar treatment of SD and SY were 34 and 32, respectively (Figure 5B). This indicates that biochar can increase the number of unique OTUs of bacteria and fungi in soil and can alter the composition of bacteria and fungi groups. This is due to the ability of biochar to affect soil physical and chemical properties and biological characteristics to thus create a specific microenvironment for microorganisms [63].

### 2.6. Effects of Biochar on the Interaction of Soil Bacteria and Fungi

In this study, the co-expression analysis based on Python plotting revealed the top 50 different genera with the highest correlation, and these are presented in Figure 6 and Figure 7. The horizontal correlation network diagram of genera revealed that there is significant interaction between different genera (the orange line represents positive correlation, the green line represents negative correlation, the thickness of the line represents the magnitude of the correlation coefficient, and the number of lines represents the close degree of connection between nodes). Figure 6 shows that in the interaction network of major soil bacteria genera, the high relative abundance microorganisms of SBCK were *Candidatus_Solibacter*, *Uncultured_Bacterium_F_Anaerolineaceae*, and *Uncultured_Bacterium_F_Da101_Soil_Group*. The groups with higher relative abundance in SD were *uncultured_Bacterium_F_Acidobacteriaceae_ [Subgroup_1]* and *Uncultured_Bacterium_F_Da101_Soil_Group*. Those with high relative abundance in SY were *Uncultured_Bacterium_F_Acidobacteriaceae_[Subgroup_1]*, *Uncultured_Bacterium_F_Da101_Soil_Group* and *Anaeromyxobacte R*. The degrees of correlation among the bacterial genera were different. Among the SBCK, 38 were negatively correlated, and 29 were positively correlated. Among the SD treatments, 48 were negatively correlated, and 34 were positively correlated. In the SY treatment group, 48 were negatively correlated, and 31 were positively correlated. It can be observed that the correlation of major soil bacteria genera in the biochar treatment group is stronger than SBCK, and the number of positive correlations (orange) in the biochar treatment group was higher than SBCK.

As presented in Figure 7, in the interaction network of major soil fungi genera, *Pseudeurotium* exhibited the highest relative abundance in SBCK, while *Cladosporium*, *Aspergillus,* and *Alternaria* exhibited the highest relative abundance in SD. The high relative abundances in SY were *Pseudeurotium*, *Didymella*, *Alternaria,* and *Solicoccozyma*. The degrees of correlation among the fungal genera were different. Among SBCK, 39 fungal genera were negatively correlated, and 30 fungal genera were positively correlated. There were 47 negative correlations and 41 positive correlations among SD treatments. In the SY treatment group, 43 exhibited a negative correlation, and 35 possessed a positive correlation. It can be observed that the correlation of soil fungal genera in the biochar treatment group was stronger than SBCK, and the number of positive correlations (orange) in the biochar treatment group was significantly higher than that in the SBCK group. Biochar increases the interaction level between microorganisms, and this may affect the complexity of the network structure. This indicates that biochar application can not only alter the microbial community structure but also change the overall microbial interaction.

### 2.7. Correlation Analysis of Soil Microbial Community and Nutrients

Soil physical and chemical properties such as pH, water content, organic matter, and soil nutrients are important factors that affect the soil microbial community [24]. As shown in Figure 8, the RDA results in this study revealed that pH, organic matter, AN, AP, and AK all exerted a significant influence on soil bacteria and fungi and exhibited a significant correlation, and these results were similar to those of previous studies [64,65]. pH exerts a large influence on the suitable living environment for microorganisms [66]. Organic matter content (OC) is the most important measurement index of soil nutrients, and nitrogen is an essential nutrient for microorganisms [67].

Redundancy analysis revealed that the main bacterial populations in the soil were correlated with the soil nutrients (Figure 8A). The relationship between rays in the figure is represented by the included angles, with obtuse angles representing negative correlation and acute angles representing positive correlation. The RDA analysis revealed that the eigenvalues of the two main axes were 25.74% and 20.95%, respectively. Among these, TK, OC, and AN were positively correlated with *Actinobacteria* and were distributed in the first quadrant. TK, OC, and AN were negatively correlated with *Ignavibacteriae*, *Bacteroidetes,* and *Verrucomicrobia*, and were concentrated in the third quadrant. AK was positively correlated with *Chloroflexi* and distributed in the fourth quadrant, while pH was positively correlated with *Acidobacteria* and *Armatimonadetes* and distributed in the second quadrant. Soil TK was the environmental factor that possessed the highest explanatory degree.

The major fungal populations in the soil were correlated with soil nutrients (Figure 8B). The RDA analysis revealed that the two main axis eigenvalues were 33.4% and 19.16%, respectively. AK was positively correlated with *Rozellomycota* and *Chytridiomycota* and was distributed in the first quadrant. AK was negatively correlated with *Mortierellomycota* and *Cercozoa*, which were distributed in the third quadrant. AP, AN, and OC were positively correlated with *Aphelidiomycota* and were distributed in the fourth quadrant. pH was positively correlated with *Mortierellomycota*, *Ascomycota,* and *Basidiomycota,* which were distributed in the second quadrant.

The results showed that the composition of soil bacterial and fungal communities was closely related to soil nutrient properties, such as soil pH, AN, AP, and AK, suggesting that the effect of biochar on soil community structure might be indirectly driven by the change in soil properties.

## 3. Materials and Methods

### 3.1. Test Site

The experiment was performed at the 26°10′ N, 119°23′ E pot plant test base at Heilongjiang Bayi Agricultural University. The environmental parameters included an annual sunshine time of 2726 h, a frost-free period of 166 d, an annual average temperature of 4.2 °C, an annual precipitation of 427 mm, and an annual evaporation of 1635 mm.

### 3.2. Test Materials

The test soil type was Northeast Meadow albic soil that was acquired from 850 Farm Science and Technology Park, Hulin City, Heilongjiang Province, China. The background values of soil basic nutrients (0–20 cm plow layer) included a soil organic matter content of 34.80 g/kg, a soil total nitrogen content of 1.70 g/kg, a soil total phosphorus content of 0.88 g/kg, an alkali hydrolyzable nitrogen content of 162.00 mg/kg, an available phosphorus content of 45.30 mg/kg, an available potassium content of 97.00 mg/kg, a pH value of 5.35, and a CEC value of 10.16 cmol/kg. Biochar was provided by Liaoning Jinhefu Co. Ltd., Liaoning, China. The total C content of corn stalk biochar was 50.60%, the total N content was 1.4%, the ash content was 15.34%, and the pH value was 8.50. The total C content, total N content, ash content, and pH value of rice straw biochar were 51.41%, 1.45%, 23.89%, and 7.87, respectively. The rice variety was Kenjing 5 and was provided by the Rice Center of Heilongjiang Bayi Agricultural University.

### 3.3. Experimental Design

The pot experiment was conducted using a pot with a diameter of 30 cm and a height of 28 cm. Rice straw and corn straw biochar were selected, and the treatments included SBCK (conventional control treatment with no biochar), SD (rice straw biochar 20 t/ha), and SY (corn straw biochar 20 t/ha) with a completely random design. Each treatment was set for 3 replicates, and each replicate comprised 20 pots. The weight of albic soil of each pot was 10 kg. The application dosages were 131.4 kg N/ha with urea and diammonium phosphate, 69.0 kg P_2_O_5_/ha with diammonium phosphate, and 78.0 kg_2_O/ha with potassium sulfate.

Urea was applied five times in the form of basal fertilizer, tillering fertilizer, regulating fertilizer, panicle fertilizer, and grain fertilizer at compositions of 30%, 30%, 10%, 20%, and 10%, respectively. Phosphate fertilizer was used for all base fertilizer applications. Potassium fertilizer was applied twice as basal fertilizer and panicle fertilizer at compositions of 60% and 40%, respectively. Biochar and albic soils were thoroughly mixed. Three holes of rice were planted in each pot, and three seedlings were planted in each hole. The rest of the experiments were performed according to the conventional management measures of rice, weeds, insects, and diseases, which were controlled by either chemical or manual methods to avoid yield loss.

### 3.4. Sample Collection

Rice rhizosphere soil with a depth of 0–20 cm was collected at the rice maturity stage by a stainless steel soil drill with a diameter of 2 cm, and 10 points were randomly selected from each treatment. After removing the roots, weeds, soil animals, and other impurities, they were mixed and used as a repeated soil sample for the same treatment. The soil sample was put into aseptic sealed bag and temporarily stored in a low-temperature ice box and was taken back to the laboratory. The soil sample was divided into two parts after being screened by 2 mm. Part of the soil samples was frozen at −40 °C for analysis of microbial bacteria (16S rRNA) and fungal (18S rRNA) communities, the other part of the soil was air-dried for the analysis of soil chemical properties.

### 3.5. Test Methods

Soil samples (0.5 g) were weighed, and DNA was extracted using the Mobo Power Soil DNA Isolation Kit. The purity, concentration, and integrity of the DNA were detected by agarose gel electrophoresis and spectrophotometry. The V3 + V4 region of bacterial 16SRRNA was amplified using the following primers: 338F: 5’-ACTCCTACGGGAGCAGCA-3’ and 806R: 5’-GGACTACHVGGGTWTCTAAT-3’. The ITS1 region of the fungus was amplified using the following primers: 5’-CTTGGTCATTTAGAGGAAGTAA-3’ and 5’-GCTGCGTTCTTCATCGATGC-3’. The PCR amplification reaction volume was 50 µL. The reaction procedure included pre-denaturation at 95 °C for 5 min, 35 cycles of 95 °C for 30 s, 50 °C for 30 s, and 72 °C for 40s, and then 72 °C for 7 min. After amplification, the original library of the samples was established and sequenced using the Illumina HiSeq 2500 platform (Illumina Corporation, San Diego, CA, USA) with a 2 × 250 bp double-ended sequencing strategy. The original data were stitched (Flash, version 1.2.11), and the stitched sequences were filtered according to quality (Trimmomatic, version 0.33). The UCHIME (version 8.1) was removed to obtain the tag sequence with high quality. When the sample sequencing depth index value was greater than 99%, the sequencing data were considered to be reasonable. The extraction and sequencing of soil microbial total DNA were performed by Beijing Biamark Biotechnology Co. Ltd., Beijing, China [48,68].

Soil pH was measured in 1:2.5 ratio soil solutions (with de-ionized water) with a pH meter. The organic matter content was measured using the high temperature-volume method, with heating and oxidation by potassium dichromate. For total nitrogen, H_2_SO_4_ was used as an accelerator for digestion, and then the Kjeldahl analytic method was used. The soil alkali-hydrolyzable nitrogen was measured with the alkaline hydrolysis diffusion method. Available phosphorus was extracted with sodium bicarbonate and determined with ultraviolet spectrophotometry (TU-1810, Beijing Pgeneral Instrument Co. Ltd., Beijing, China). Total phosphorus was measured using the alkali fusion-molybdenum antimony anti-spectrophotometric method. Soil total potassium (TK) and available potassium (AK) were quantified using inductively coupled plasma-atomic emission spectrometry (ICPS-7500, Shimadzu, Japan). All the above chemical indexes were measured according to Soil Agrochemical Analysis, which was published by China Agriculture Press [69].

### 3.6. Data Processing and Analysis

OTUs exhibiting a similarity of 97% were randomly selected to generate dilution curves, and the richness indices for Chao and Ace, Simpson, and Shannon were calculated using Mothur software (version 1.31.2). The OTU was annotated based on the RDP and Unite taxonomic databases, and Excel and SPSS were used for data processing. Excel and R language tools were used to draw histograms and Veen plots for the statistical results of species composition and relative abundance of the samples.

## 4. Conclusions

The results of this study revealed that the biochar increased the soil nutrient content of albic soil, and biochar addition increased soil bacteria and fungi abundance and altered community structure. The bacteria ACE indexes of SD and SY were increased by 3.10% and 2.06%, respectively, and the fungi ACE and Chao indices of SD were increased by 7.86% and 14.16%, respectively, compared to that of SBCK. Biochar increased the number of unique OTUs of bacteria and fungi in the soil. The relationship between soil bacteria and fungi in the biochar treatments was stronger than SBCK. In addition, the changes in the soil bacteria and fungi community compositions were closely related to soil nutrient characteristics, such as pH, OC, AN, AP, and AK, and these characteristics were correlated with biochar addition, which suggested that the impacts of biochar on the soil bacteria and fungi community were indirectly driven by alternation of soil nutrient characteristics. The addition of two types of biochar altered the soil microbial community structure, and the effect of rice straw biochar treatment was more pronounced. These results aimed to provide a reference and basic understanding for albic soil improvement by biochar, with good application prospects.

## Figures and Tables

**Figure 1 molecules-26-04783-f001:**
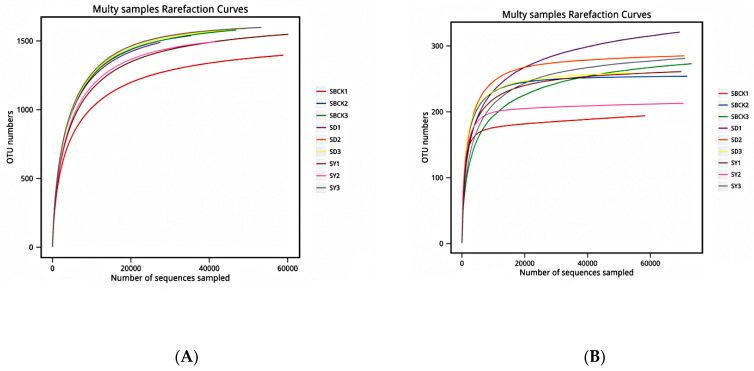
Bacterial (**A**) and fungal (**B**) rarefaction curves from different soil samples (SBCK: conventional control treatment with no biochar; SD: rice straw biochar; SY: corn straw biochar). Note: OUT: operational taxonomic units.

**Figure 2 molecules-26-04783-f002:**
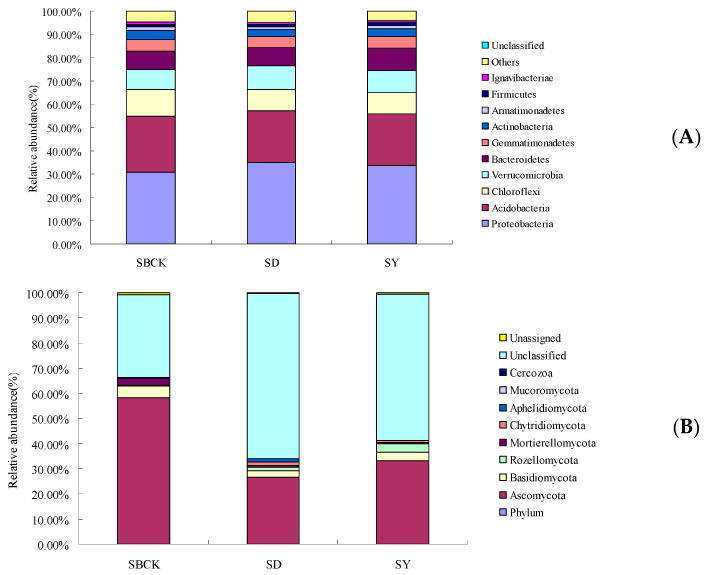
Changes in the relative abundance of soil bacterial species at phylum level (**A**) and fungal species at phylum (**B**) level with biochar applied (SBCK: conventional control treatment with no biochar; SD: rice straw biochar; SY: corn straw biochar).

**Figure 3 molecules-26-04783-f003:**
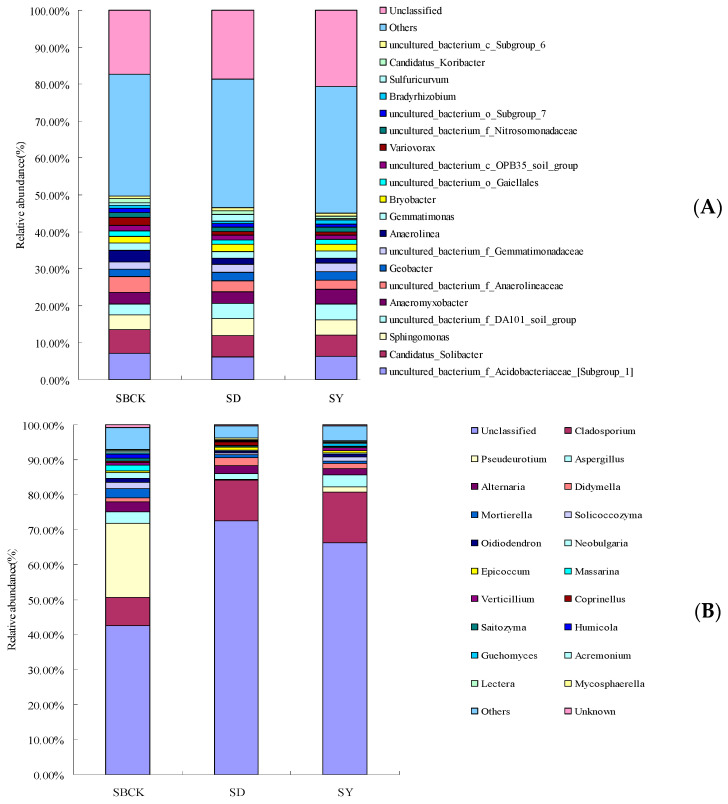
Changes in relative abundance of soil bacterial species at the genus level (**A**) and fungal species at genus level (**B**) with biochar applied (SBCK: conventional control treatment with no biochar; SD: rice straw biochar; SY: corn straw biochar).

**Figure 4 molecules-26-04783-f004:**
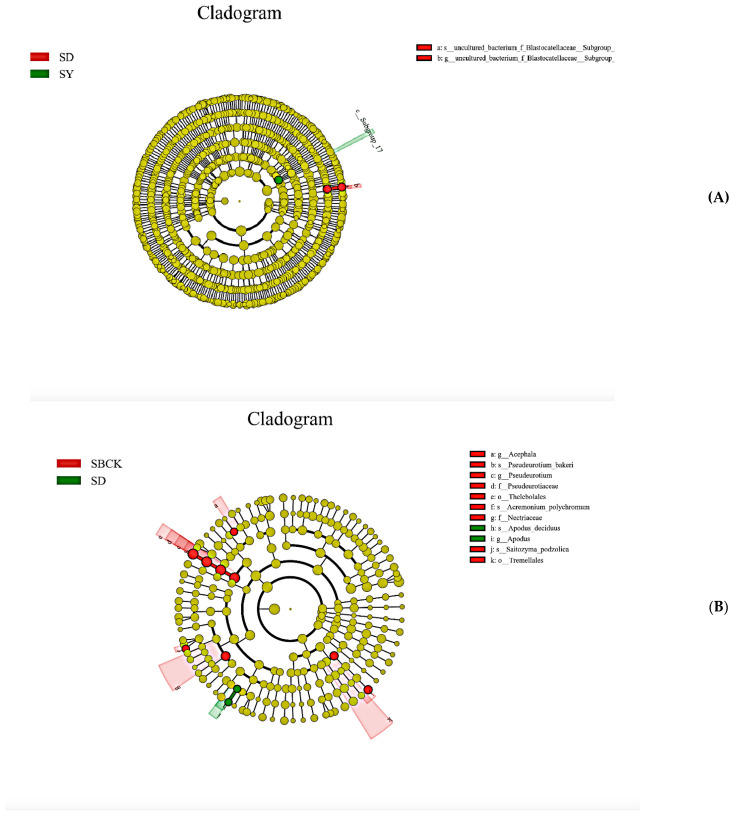
Significance analysis of differences among different treatment groups. Note: (**A**) soil bacteria, (**B**) soil fungi (SBCK: conventional control treatment with no biochar; SD: rice straw biochar; SY: corn straw biochar).

**Figure 5 molecules-26-04783-f005:**
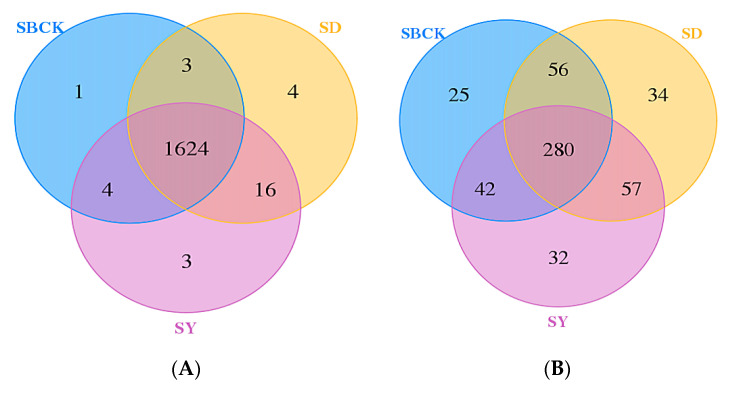
Venn diagrams of soil bacterial (**A**) and fungal (**B**) communities in soil (SBCK: conventional control treatment with no biochar; SD: rice straw biochar; SY: corn straw biochar).

**Figure 6 molecules-26-04783-f006:**
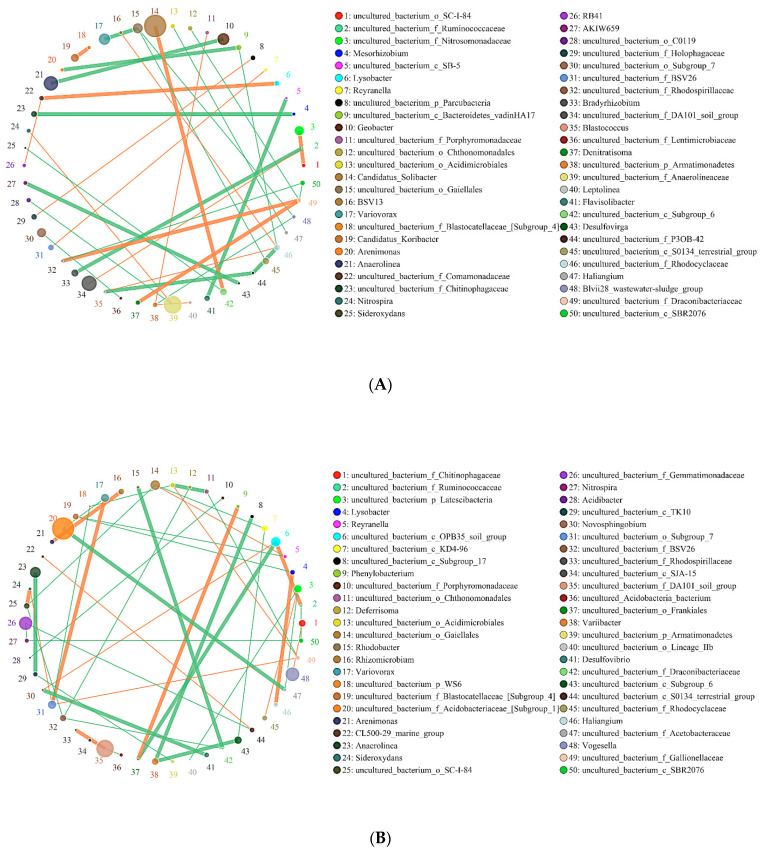
Network of the interactions of the main bacteria genus in soil. Note: (**A**) SBCK (conventional control treatment with no biochar); (**B**) SD (rice straw biochar); (**C**) SY (corn straw biochar).

**Figure 7 molecules-26-04783-f007:**
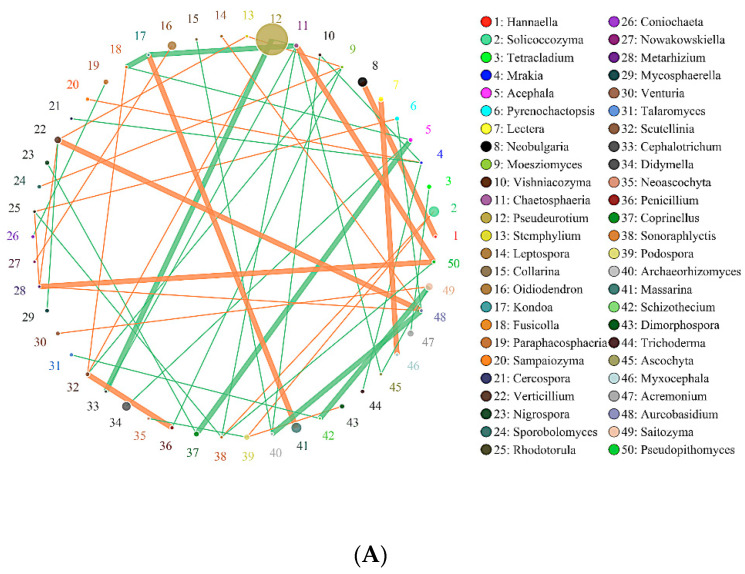
Network of interactions of the main fungi genus in soil. Note: (**A**) SBCK (conventional control treatment with no biochar); (**B**)SD: (rice straw biochar); (**C**) SY: (corn straw biochar).

**Figure 8 molecules-26-04783-f008:**
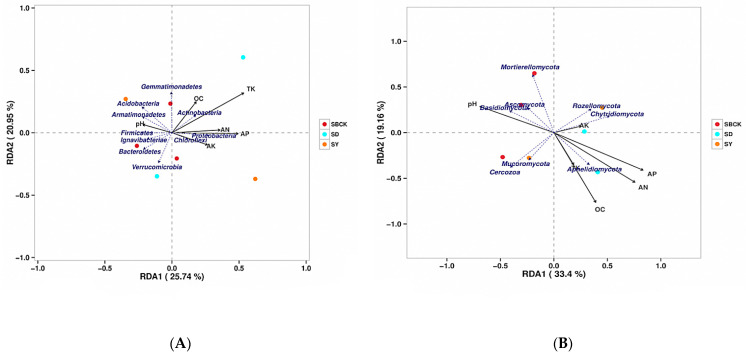
RDA analysis of the effects of soil nutrients on changes in bacterial community (**A**) and fungal community (**B**) structure. Note: SBCK (conventional control treatment with no biochar); SD (rice straw biochar); SY: (corn straw biochar); TK: total potassium; OC: organic matter content; AN: available nitrogen; AP: available phosphorus; AK: available potassium.

**Table 1 molecules-26-04783-t001:** Effects of biochar on the soil nutrient content in the rice maturation stage.

	pH	Organic Matter (OC) (g/kg)	Alkali-Hydrolyzed Nitrogen (AN) (mg/kg)	Available Phosphorus (AP) (mg/kg)	Available Potassium (AK) (mg/kg)	Total Nitrogen (TN) (g/kg)	Total Phosphorus (TP) (g/kg)	Total Potassium (TK) (g/kg)
SBCK	6.80 ^a^	38.27 ^a^	147.67 ^c^	57.20 ^b^	121.00 ^b^	1.85 ^a^	1.04 ^a^	17.33 ^a^
SD	6.30 ^a^	38.67 ^a^	166.67 ^a^	62.47 ^a^	126.00 ^a^	1.94 ^a^	1.08 ^a^	17.40 ^a^
SY	6.60 ^a^	38.77 ^a^	158.67 ^b^	62.13 ^a^	104.67 ^c^	1.93 ^a^	1.13 ^a^	18.27 ^a^

Note: SBCK: conventional control treatment with no biochar; SD: rice straw biochar; SY: corn straw biochar; ^a–c^ Columns reporting different letters are significantly different at *p* < 0.05 (*n* = 3, LSD test).

**Table 2 molecules-26-04783-t002:** The abundance and diversity of OTUs from soil samples where biochar was applied.

Treatment	ACE Index	Chao Index	Simpson Index	Shannon Index	ACE Index	Chao Index	Simpson Index	Shannon Index
	Bacterial (V3 + V4)				Fungal (ITS1)			
SBCK	1555.28 ^a^	1569.04 ^a^	0.01 ^a^	6.02 ^a^	278.18 ^ab^	269.35 ^ab^	0.12 ^ab^	3.34 ^a,b^
SD	1603.53 ^a^	1619.97 ^a^	0.01 ^a^	6.05 ^a^	300.04 ^a^	307.49 ^a^	0.08 ^b^	3.47 ^a^
SY	1587.25 ^a^	1604.48 ^a^	0.01 ^a^	6.00 ^a^	262.53 ^b^	262.74 ^b^	0.14 ^a^	3.11 ^b^

Note: SBCK: conventional control treatment with no biochar; SD: rice straw biochar; SY: corn straw biochar; ^a–b^: columns reporting different letters are significantly different at *p* < 0.05 (*n* = 3, LSD test); ITS1: internal transcribed spacer-1 region.

**Table 3 molecules-26-04783-t003:** Reads for observed soil bacterial and fungal OTUs.

Treatment	Bacterial (V3 + V4) OTU Numbers	Coverage	Fungal (ITS1) OTU Numbers	Coverage
SBCK	1504.33 ^a^	0.9973 ^a^	240.33 ^b^	0.9998 ^a^
SD	1549.67 ^a^	0.9961 ^a^	288.33 ^a^	0.9997 ^a^
SY	1546.67 ^a^	0.9979 ^a^	251.67 ^a,b^	0.9998 ^a^

Note: SBCK: conventional control treatment with no biochar; SD: rice straw biochar; SY: corn straw biochar; OUT: operational taxonomic units; ^a–b^: columns reporting different letters are significantly different at *p* < 0.05 (*n* = 3, LSD test); ITS1: internal transcribed spacer-1 region.

## Data Availability

All data, models, and code generated or used during the study appear in the submitted article.

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
