# Peer review of "Impact of Different Biochars on Microbial Community Structure in the Rhizospheric Soil of Rice Grown in Albic Soil"

_molecules, 2021, doi:10.3390/molecules26164783_

Round 1
Reviewer 1 Report
The MS molecules-1298410 with the title (Effects of biochar on soil bacterial and fungal community structure in the rice root zone of albic soil) was carried out to to clarify the effects of biochar on the diversity of bacteria and fungi in the rice root zone and to reveal the changes in soil microbial community structure in the root zone after biochar application to provide a scientific basis for the improvement of albic soil. Below are my comments:
The Affiliations are not written according to the format of the journal.
Why the authors choose these two feedstocks of biochar (Rice and corn stalk biochar)?
The authors must write the full words of all abbreviations before the first mention, there are many abbreviations without writing the full words in first mention such as SD, SY, OUT, SBCK, CK, etc. Please check this in the whole MS.
L31-32 Why capital letters >> Proteobacteria, Verrucomicrobia, and Bacteroidetes>> Rozellomycota, Chytridiomycota, and Apheliidiomycota
L48-51 >> Who said this? Please cite a reference
L 51>> what is this hm2? Something is wrong here
L57-58>> Who said this>> The microbial community structure is closely related to soil quality and is extremely sensitive to changes in the living environment?? Please cite the reference here and where are possible in the MS>>
Hewedy, O.A.; Abdel Lateif, K.S.; Seleiman, M.F.; Shami, A.; Albarakaty, F.M.; M. El-Meihy, R. Phylogenetic Diversity of Trichoderma Strains and Their Antagonistic Potential against Soil-Borne Pathogens under Stress Conditions. Biology 2020, 9, 189.
L61-76 In this paragraph, every sentence you start with Biochar >> Biochar >> Biochar>> try to use It (means biochar also).
L67 >> Who said this>> Biochar can promote the growth of fungi and the colonization of the root system of crops? Please cite this recent publication here and where possible in MS>> Alkharabsheh, H.M.; Seleiman, M.F.; Battaglia, M.L.; Shami, A.; Jalal, R.S.; Alhammad, B.A.; Almutairi, K.F.; Al-Saif, A.M. Biochar and Its Broad Impacts in Soil Quality and Fertility, Nutrient Leaching and Crop Productivity: A Review. Agronomy 2021, 11, 993.
L77-80>> put the aim of the study in an independent paragraph, and try to add also the treatments here.
I recommend authors to add some sentences about the biochar made from rice or corn in the paragraph of biochar in the introduction section. Authors can look at>> Seleiman, M.F.; Alotaibi, M.A.; Alhammad, B.A.; Alharbi, B.M.; Refay, Y.; Badawy, S.A. Effects of ZnO Nanoparticles and Biochar of Rice Straw and Cow Manure on Characteristics of Contaminated Soil and Sunflower Productivity, Oil Quality, and Heavy Metals Uptake. Agronomy 2020, 10, 790.
Results and discussion:
The figures are not clear, the resolution is not good quality.
In table, authors must write below the table and figures the meaning of all abbreviations.
In tables, the authors added characters beside the values, and I guess that these characters are for the significance. I noted that the authors used only one character (a), this means that there is not significant differences between the treatments but I do not think so according to the differences between the values in some measurements.
The discussion part should be improved
I was looking for the plant growth and yield measurments, did the authors measure these parameters?? Because the author had a pot experiment and had the plants until maturity as they mention
Material and methods:
L352>> The rice variety is Kenjing 5>> use WAS instead of IS
L358> what is this hm2? Ha or m2??? Can not be hm2?? Check this in whole ms, I found same mistake in othere places
Chemical fertilizers values were based on the product or the raw element? I mean for example 225 kg N or kg urea??? The question is also for P and K elements??
The format of the reference is not according to the guidelines of the authors for this journals>> Something should be BOLD and other things should be ITALICS, etc.
Reviewer 2 Report
Comment 1: Please write abbreviations for ITS, SD, SY, SBCK and so on both in the abstract and the main text.
Comment 2: Please reduce the number of words or sentences in abstract and rewrite.
Comment 3: Write key words as per authors instructions.
Comment 4: Please include more information and references in the introduction and rewrite.
Comment 5: There are lot of information in the results and discussion. But did not much discussion with previous references. Please discuss in details.
Comment 6: For line space follow author instructions.
Comment 7: Reduce the number of sentences in the conclusion and rewrite.
Author Response
Response to Editor and Reviewers
Dear Editors and reviewers,
Thank you for your letter and for the reviewers’ comments concerning our manuscript entitled “Effects of biochar on soil bacterial and fungal community structure in the rice root zone of albic soil ”(molecules-1298410). The comments are all valuable and very helpful for revising and improving our paper, as well as the important guiding significance to our researches. We have studied comments carefully and have made correction which we hope meet with approval. Revised portion are marked in green in the paper. The main corrections in the paper and the responds to advices are as follow:
Comment 1.Please write abbreviations for ITS, SD, SY, SBCK and so on both in the abstract and the main text.
Reply: Thanks a lot for the valuable advice, and the author made modifications in accordance with expert advice. The all abbreviations have been written in full when first mentioned, including SD, SY, SBKC, OUT and ITS 1, both in the abstract and the main text.
Comment 2. Please reduce the number of words or sentences in abstract and rewrite.
Reply: Thanks a lot for the valuable advice. The author has integrated and optimized the abstract, and added a summary statement at the end of the abstract, as well as the future development prospect.
Comment 3.Write key words as per authors instructions.
Reply: Thanks for the valuable advice, and the key words have been modified.
Comment 4.Please include more information and references in the introduction and rewrite.
Reply: Thanks a lot for for the valuable advice. The introduction was carefully revised, including the general situation and existing problems of albic soil, the role of microorganisms, Plant growth-promoting rhizobactria (PGPR), the characteristics of biochar, and the entry point of the article. And references from 2020 to 2021 were added.
Comment 5.There are lot of information in the results and discussion. But did not much discussion with previous references. Please discuss in details.
Reply: Thanks a lot for for the valuable advice. The results and discussion were carefully analyzed, including soil nutrients, microbial richness and diversity, and microbial community structure, especially at the phylum and genus level. In the analysis of key indicators, the previous research progress is expounded, and the reasons for the results of this test are carefully analyzed in details.
Comment 6.For line space follow author instructions.
Reply: Thanks for the valuable advice, and the line space follow author instructions, and the Other formats have also been modified, such as references. In addition, the author revises all the images in the paper to enhance the resolution, Changes have been made from Fig.1 through Fig.8.
Comment 7.Reduce the number of sentences in the conclusion and rewrite.
Reply: Thanks for the valuable advice, The author reduced the number of conclusions, re-integrated and optimized the conclusions, and at the end of the conclusion added the future development prospects of the technology.
We have also checked other sections of the discussion to ensure the accuracy of our description. We tried our best to improve the manuscript and made some changes in the manuscript. We appreciate for Editors/Reviewers’ warm work earnestly and hope that the correction will meet with approval. The revised manuscript has been resubmitted to the journal. We hope it can be able to achieve published standards.
Sincerely,
Dawei Yin and Liang Jin
E-mail: yindazhiyindawei@126.com

Reviewer 3 Report
Comments on molecules-1298410
Title
It may be modified as “Impact of different biochars on microbial community structure in the rhizospheric soil of rice grown in albic soil”
Abstract
Line 28-31, the sentence may be modified as “The results revealed that the ACE indexes of SD and SY were increased by 3.10% and 2.06%, respectively, and that the Chao indexes of SD and SY were increased by 3.25% and 2.26%, respectively, compared to that of SBCK.”
Line 35-36, “Biochar increases the number of unique OTUs of bacteria and fungi in the soil and alters the composition of the bacterial and fungal groups.” Is it fact or your finding? If finding, write in the past tense
Line 42, The word “PH” may be corrected “pH” and crosschecked throughout the manuscript
A clear conclusion is missing
Only the main and the most prominent results should be presented here in the abstract
Also, there is no description of the treatments applied
Also, a sentence regarding the importance of the conducted research work
Introduction
The introduction is very short. There should be more focus on the role of microorganisms in improving soil fertility like plant growth-promoting rhizobacteria.
The Introduction section needs to be updated with the latest papers from 2020-21. I could not find any from 2020-2021. The following papers may be considered to improve the discussion section
https://doi.org/10.1007/s42729-021-00514-z
https://doi.org/10.1016/j.geoderma.2020.114803
https://doi.org/10.1111/jac.12502
Results and Discussion
This section needs to be thoroughly revised. It has a major issue regarding presenting the results and graphical representation
The treatments have been described wrongly. Their description is different from that discussed in Materials and Methods
Figure 2, 3, 4 and 6, needs to be redrawn, as it is very difficult to read
Same with Figure 8, improve visibility
Regarding the use of different biochars, and focus on microbial community structure, it would be better to present the surface structure before and after the application as visualized through SEM
The discussions presented regarding the results obtained are very shallow. It needs more deep explanations regarding different mechanisms behind the improvement in different parameters. Moreover, the Discussion section needs to be updated with the latest papers from 2020-21.
Materials and Methods
What is “hm2”. The application rates should be as per SI units
How much soil was taken in each pot?
How many replications were applied?
What conventional practices were applied line 268=269, give an explanation
The method to sample rhizospheric soil is not clear. It needs more explanation and clarification
In “Test methods”, no reference was used
Conclusion
The treatment description is different from materials and Methods
The overall conclusion is missing at the end
One sentence for future research work needed may be added at the end of this section
Author Response
Response to Editor and Reviewers
Dear Editors and reviewers,
Thank you for your letter and for the reviewers’ comments concerning our manuscript entitled “Effects of biochar on soil bacterial and fungal community structure in the rice root zone of albic soil ”(molecules-1298410). The comments are all valuable and very helpful for revising and improving our paper, as well as the important guiding significance to our researches. We have studied comments carefully and have made correction which we hope meet with approval. Revised portion are marked in green in the paper. The main corrections in the paper and the responds to advices are as follow:
Comments on molecules-1298410
Comment 1.Title
It may be modified as “Impact of different biochars on microbial community structure in the rhizospheric soil of rice grown in albic soil”
Reply:Thanks a lot for for the valuable advice, and the author made modifications in accordance with expert advice. The article uses the topic of experts advice. for “Impact of different biochars on microbial community structure in the rhizospheric soil of rice grown in albic soil”.
Comment 2.Abstract
Line 28-31, the sentence may be modified as “The results revealed that the ACE indexes of SD and SY were increased by 3.10% and 2.06%, respectively, and that the Chao indexes of SD and SY were increased by 3.25% and 2.26%, respectively, compared to that of SBCK.”
Reply:Thanks a lot for for the valuable advice, and the sentence had modify as “Compared to SBCK, the results revealed the ACE indices of SD and SY of bacteria were increased by 3.10% and 2.06%, and the Chao indexes of SD and SY of bacteria were increased by 3.25% and 2.26%, respectively”.
Comment 3.Line 35-36, “Biochar increases the number of unique OTUs of bacteria and fungi in the soil and alters the composition of the bacterial and fungal groups.” Is it fact or your finding? If finding, write in the past tense
Reply:Thanks to the expert's advice, and the author changed this sentence to the past tense. For “This indicates that biochar could increase the number of unique OTUs of bacteria and fungi in soil and could alter the composition of bacteria and fungi groups” . In 383-385 lines.
Comment 4.Line 42, The word “PH” may be corrected “pH” and crosschecked throughout the manuscript
Reply:Thanks to the expert's advice, and the author has revised the writing method of the pH value of the whole paper.
Comment 5.A clear conclusion is missing
Reply:Thanks a lot for for the valuable advice, and the author has carefully revised and summarized the conclusion of the article, In 583-594 lines. And as shown below:
The results of this study revealed that the biochar increased the soil nutrient content of albic soil, and biochar addition increased soil bacteria and fungi abundance and altered community structure .The bacteria ACE indexes of SD and SY were increased by 3.10% and 2.06%, respectively, and the fungi ACE and Chao indices of SD were increased by 7.86% and 14.16%, respectively, compared to that of SBCK. Biochar increased the number of unique OTUs of bacteria and fungi in the soil. The relationship between soil bacteria and fungi in the biochar treatments was stronger than SBCK. In addition, the changes of the soil bacteria and fungi community compositions were closely related to soil nutrient characteristics, such as pH, OC, AN, AP and AK, while these characteristics were correlated with biochar addition, which suggested that the impacts of biochar on the soil bacteria and fungi community were indirectly driven by alternation of soil nutrient characteristics. The addition of two types of biochar altered the soil microbial community structure, and the effect of rice straw biochar treatment on SD was more pronounced.The results obtained in this study indicated that the biochar application to soil provides a new approach to improve albic soil, with good application prospects.
Comment 6.Only the main and the most prominent results should be presented here in the abstract
Reply:Thanks a lot for for the valuable advice, the author has made a comprehensive revision to the abstract, In 583-594 lines.
Comment 7.Also, there is no description of the treatments applied
Reply:Thanks for the valuable advice. The description of the treatment has been carefully modified by the author, In 516-522 lines. And as shown below:
The pot experiment was conducted using a pot with a diameter of 30 cm and a height of 28 cm. Rice straw and corn straw biochar were selected, and the treatments included SBCK (conventional control treatment with no biochar), SD (rice straw biochar 20t/ha), and SY (corn straw biochar 20t/ha) with a completely random design. Each treatment was set for 3 replicates, and each replicate were 20 pots. The application dosages were 131.4 kg N/ha with urea and diammonium phosphate, 69.0 kg P2O5/ha with diammonium phosphate and 78.0 kg2O/ha with potassium sulfate.
Comment 8.Also, a sentence regarding the importance of the conducted research work
Reply:Thanks for the valuable advice. The article is added at the end of both the abstract and the conclusion, for “The results obtained in this study indicated that the biochar application to soil provides a new approach to improve albic soil, with good application prospects.”
Comment 9.Introduction
The introduction is very short. There should be more focus on the role of microorganisms in improving soil fertility like plant growth-promoting rhizobacteria.
Reply:Thanks a lot for for the valuable advice. The introduction was carefully revised, including the general situation and existing problems of albic soil, the role of microorganisms on the role of microorganisms in improving soil fertility, Plant growth-promoting rhizobactria (PGPR), the characteristics of biochar, and the entry point of the article. And references from 2020 to 2021 were added.
Comment 10.The Introduction section needs to be updated with the latest papers from 2020-21. I could not find any from 2020-2021. The following papers may be considered to improve the discussion section
https://doi.org/10.1007/s42729-021-00514-z
https://doi.org/10.1016/j.geoderma.2020.114803
https://doi.org/10.1111/jac.12502
Reply:Thanks a lot for for the valuable advice. The Introduction section was cited with the latest papers from 2020-2021, and the author makes a detailed analysis of the important indicators. including soil nutrients, microbial richness and diversity, and microbial community structure, especially at the phylum and genus level. In the analysis of key indicators, the previous research progress is expounded, and the reasons for the results of this test are carefully analyzed in details. And It also cites the latest research results of biochar as below:
13. Rizwan, M. S.;Imtiaz, M.;Zhu, J.;Yousaf, B.;Hu, H. Immobilization of pb and cu by organic and inorganic amendments in contaminated soil.Geoderma,2021,385(4), 114803.
17. Ullah, N.;Ditta, A.;Imtiaz, M.;Li, X.M.; Jan,A.U.;Mehmood,S.;Rizwan,M.S.;Rizwan,M. Appraisal for organic amendments and plant growth-promoting rhizobacteria to enhance crop productivity under drought stress: a review. J.Agron.Crop.Sci. 2021,00,1-20.
53. Ullah, N.;Ditta, A.;Khalid, A.;Mehmood, S.;Rizwan, M.S.;Ashraf, M.;Mubeen, F.;Imtiaz, M.;Iqbal, M.M. Integrated effect of algal biochar and plant growth promoting rhizobacteria on physiology and growth of maize under water deficit irrigations.J.Soil.Sci.Plant.Nut. 2020, 20, 346–356.
Comment 11.Results and Discussion
This section needs to be thoroughly revised. It has a major issue regarding presenting the results and graphical representation
Reply:Thanks for the valuable advice. The author carefully revised the results and discussion section, and the author revised all the images in the paper to enhance the resolution. Changes have been made from Fig.1 through Fig.8.
Comment 12..The treatments have been described wrongly. Their description is different from that discussed in Materials and Methods
Reply:Thanks a lot for for the valuable advice. The author revises the treatment number and description in the result and discussion section and the material and method section. Rice straw and corn straw biochar were selected, and the treatments included SBCK (conventional control treatment with no biochar), SD (rice straw biochar 20t/ha), and SY (corn straw biochar 20t/ha).
Comment 13.Figure 2, 3, 4 and 6, needs to be redrawn, as it is very difficult to read
Reply:Thanks for the valuable advice.The author revised all the images in the paper to enhance the resolution. Changes have been made from Fig.1 through Fig.8.
Comment 14.Same with Figure 8, improve visibility
Reply:Thanks for the valuable advice. The author revised all the images in the paper, including Fig.8.
Comment 15.Regarding the use of different biochars, and focus on microbial community structure, it would be better to present the surface structure before and after the application as visualized through SEM.
Reply:Thanks a lot for for the valuable advice. The author had use SEM to analyze the soil microstructure in past year(2017-2018), and had achieved obvious results.(Biochar as Tool to Improve Physicochemical Properties of Chinese Albic Soils, 10.1166/jbmb.2018.1735). However, in this experiment, the author did not keep fresh soil samples, and missed the opportunity to use scanning electron microscopy. We will pay attention to this link in future experiments.
Comment 16.The discussions presented regarding the results obtained are very shallow. It needs more deep explanations regarding different mechanisms behind the improvement in different parameters. Moreover, the Discussion section needs to be updated with the latest papers from 2020-21.
Reply:Thanks a lot for for the valuable advice. The results and discussion were carefully analyzed, including soil nutrients, microbial richness and diversity, and microbial community structure, especially at the phylum and genus level. In the analysis of key indicators, the previous research progress is expounded, and the reasons for the results of this test are carefully analyzed in details. At the same time, the author supplemented the relevant literature on biochar from 2020 to 2021.
Comment 17.Materials and Methods
What is “hm2”. The application rates should be as per SI units
Reply:Thanks for the valuable advice. The author made modifications in accordance with expert advice. This symbol of hm2 means hectare, that's 10,000 square meters. The hm2 in the whole text has been changed to Ha.
Comment 18.How much soil was taken in each pot?
Reply:Thanks for the valuable advice. The weight of albic soil of each pot was 10kg. And it's added in the text, lines 527-528
Comment 19.How many replications were applied?
Reply:Thanks for the valuable advice. Each treatment was set for 3 replicates, and each replicate were 20 pots.
Comment 20.What conventional practices were applied line 268=269, give an explanation
Reply:Thanks a lot for for the valuable advice. The conventional practices as following: The rest of the experiments were performed according to the conventional management measures of rice, Weeds, insects, and diseases were controlled by either chemical or manual methods to avoid yield loss. In addition, lines 537-539 in the article are added and modified.
Comment 21.The method to sample rhizospheric soil is not clear. It needs more explanation and clarification
Reply:Thanks for the valuable advice. The sampling method of the samples was carefully modified by the author, in lines 542-550. as follows:
Rice rhizosphere soil with a depth of 0-20 cm was collected at the rice maturity stage by a stainless steel soil drill with a diameter of 2 cm, and 10 points were randomly selected from each treatment. After removing the roots, weeds, soil animals and other impurities, they were mixed and used as a repeated soil sample for the same treatment. The soil sample was put into aseptic sealed bag and temporarily stored in low temperature ice box, and was taken back to the laboratory. The soil sample was divided into two parts after being screened by 2 mm.Part of the soil samples were frozen at − -40 ËšC for analysis of microbial bacteria (16S rRNA) and fungal (18S rRNA) communities, the other part of the soil was air-dried for the analysis of soil chemical properties.
Comment 22.In “Test methods”, no reference was used
Reply:Thanks for the valuable advice. The authors added references to the determination methods,in line 569-570,582.
Comment 23.Conclusion
The treatment description is different from materials and Methods
Reply:Thanks for the valuable advice. The authors have modified and unified the representation of the treatment in the material method and the conclusion, in line 591-604.
Comment 24.The overall conclusion is missing at the end
Reply:Thanks for the valuable advice. The author summarizes the conclusion of the conclusion,in lines 591-602. The main conclusion was that the impacts of biochar on the soil bacteria and fungi community were indirectly driven by alternation of soil nutrient characteristics. The addition of two types of biochar altered the soil microbial community structure, and the effect of rice straw biochar treatment was more pronounced.
Comment 25.One sentence for future research work needed may be added at the end of this section
Reply:Thanks for the valuable advice. The author added a future research work sentence at the end of the article, for “The results obtained in this study indicated that the biochar application to soil provides a new approach to improve albic soil, with good application prospects.”
We have also checked other sections of the discussion to ensure the accuracy of our description. We tried our best to improve the manuscript and made some changes in the manuscript. We appreciate for Editors/Reviewers’ warm work earnestly and hope that the correction will meet with approval. The revised manuscript has been resubmitted to the journal. We hope it can be able to achieve published standards.
Sincerely,
Dawei Yin and Liang Jin
E-mail: yindazhiyindawei@126.com

Round 2
Reviewer 1 Report
The authors should ask someone to edit the MS in terms of English. Authors should use the following citation (https://doi.org/10.1016/j.jenvman.2021.113076) instead of the reference number 25, because reference 16 and 25 in the revised version are same. Also, authors should cite reference number 14, 16, 25 in lines 108-109.
Still the Figures are not well presented, poor quality. The authors have to revise these comments before the MS can be accepted for publication in such Journal.
All authors have to read the MS and correct all typos errors.
Reviewer 2 Report
Comment 1: Please include number of replications for Tables. Make "p" become italics.
Comment 2: Please rewrite the last sentence of the abstract and conclusion.
Reviewer 3 Report
The authors have modified the manuscript as comments
Only one minor comment this time
Line 133-134, there is still BD and BY. It needs to be corrected throughout the manuscript
The visibility of almost all Figures must be improved in the final version
